# Colorectal Surgery in the COVID-19 Era: A Systematic Review and Meta-Analysis

**DOI:** 10.3390/cancers14051229

**Published:** 2022-02-27

**Authors:** Nikolaos Pararas, Anastasia Pikouli, Dimitrios Papaconstantinou, Georgios Bagias, Constantinos Nastos, Andreas Pikoulis, Dionysios Dellaportas, Panagis Lykoudis, Emmanouil Pikoulis

**Affiliations:** 3rd Department of Surgery, Attikon University Hospital, National and Kapodistrian University of Athens Medical School, Rimini 1, 12462 Chaidari, Greece; npararas@gmail.com (N.P.); anastasiapikouli@gmail.com (A.P.); georgebagias@live.com (G.B.); kosnastos@yahoo.gr (C.N.); crisismed@outlook.com (A.P.); dellapdio@gmail.com (D.D.); p.lykoudis@ucl.ac.uk (P.L.); mpikoul@med.uoa.gr (E.P.)

**Keywords:** cancer, colorectal, COVID-19, pandemic, meta-analysis

## Abstract

**Simple Summary:**

The rapid spread of the new Coronavirus-19 disease (COVID-19) has led to the implementation of unprecedented confinement measures, while healthcare systems were restructured in order to confront the pandemic; these radical measures have prevented people from seeking medical advice. At the same time, oncology and surgery societies altered treatment guidelines, favoring postponement of surgery. The aim of the present study is to determine the impact of the pandemic in the management of colorectal cancer patients. We confirmed that during the pandemic, patients were more likely to present with metastatic cancer, often requiring emergent or palliative interventions. In addition, neoadjuvant therapy and conventional open surgery utilization rates were increased in the pandemic era. These observed changes in clinical practice may be associated with tumor upstaging, which carries significant implications regarding the long-term oncologic survival of patients with colorectal neoplasias.

**Abstract:**

(1) Background: To determine the impact of the COVID-19 pandemic in the management of colorectal cancer patients requiring surgery and to examine whether the restructuring of healthcare systems led to cancer stage upshifting or adverse treatment outcomes; (2) Methods: A systematic literature search of the MedLine, Scopus, Web of Science, and CNKI databases was performed (PROSPERO ID: CRD42021288432). Data were summarized as odds ratios (OR) or weighted mean differences (WMDs) with 95% confidence intervals (95% CIs); (3) Results: Ten studies were examined, including 26,808 patients. The number of patients presenting with metastases during the pandemic was significantly increased (OR 1.65, 95% CI 1.02–2.67, *p* = 0.04), with no differences regarding the extent of the primary tumor (T) and nodal (N) status. Patients were more likely to have undergone neoadjuvant therapy (OR 1.22, 95% CI 1.09–1.37, *p* < 0.001), while emergency presentations (OR 1.74, 95% CI 1.07–2.84, *p* = 0.03) and palliative surgeries (OR 1.95, 95% CI 1.13–3.36, *p* = 0.02) were more frequent during the pandemic. There was no significant difference recorded in terms of postoperative morbidity; (4) Conclusions: Patients during the pandemic were more likely to undergo palliative interventions or receive neoadjuvant treatment.

## 1. Introduction

The rapid worldwide spread of the new SARS-CoV 19 virus led to the implementation of unprecedented confinement measures in order to minimize the dissemination of the Coronavirus-19 disease (COVID-19). At the same time, healthcare systems around the world were restructured in order to confront the pandemic; hospitals were closed or were repurposed into COVID-19 treatment centers, most prominently by suspending outpatient clinics and elective surgeries [1]. These radical measures have prevented people from carrying out annual medical screening or seeking medical advice [1,2]. Newly diagnosed cancer cases were significantly lower in 2020 compared to the pre-pandemic era [3,4], while cancer-related deaths were also significantly increased during the same time period [5].

Management of colorectal cancer, the third most common cancer worldwide [6], was consequently affected by the pandemic. The number of colonoscopies and colorectal cancer screening tests was markedly reduced in 2020 [7]; as a result, many colorectal cancer cases remained either undiagnosed or were diagnosed at an advanced stage with significant implications regarding the long-term oncologic outcomes of these patients [8]. At the same time, oncology and surgery societies altered treatment guidelines, favoring postponement of surgery [9,10]. Reports from different institutions around the world highlight the increasing number of patients presenting late, with symptoms of bowel obstruction or bowel perforation, which carry higher postoperative morbidity and mortality rates [11].

The aim of the present systematic review and meta-analysis is to determine the impact of the pandemic in the management of colorectal cancer patients requiring surgery and to assess whether the reduced accessibility to healthcare resources caused by COVID-19 led to cancer stage upshifting or adverse treatment outcomes.

## 2. Materials and Methods

A systematic literature search of the MedLine, Scopus, Web of Science, and China National Knowledge Infrastructure (CNKI) databases and clinicaltrials.gov register was conducted using a combination of the search terms “COVID-19”, “coronavirus”, “pandemic”, “colorectal cancer”, and “surgery” using the Boolean operators AND/OR as appropriate for each database. After removing duplicated studies, the titles and abstracts generated by the search algorithm were screened independently by two authors (GB, AP), and after the removal of obviously irrelevant studies, the remaining were evaluated in full-text. The reference lists were further manually checked using the snowballing technique to identify additional potentially relevant studies. Any discrepancies and disagreements ensuing during the initial screening process of the systematic literature search were resolved either by common consensus or by the mediation of a third reviewer (DD).

Studies were eligible for inclusion if they provided comparative data on tumor stage and treatment outcomes for patients managed during the pre-pandemic and pandemic time periods. The predetermined study exclusion criteria were: (1) case reports, reviews, or non-clinical studies, (2) studies published in non-English languages, (3) studies with non-surgically treated patients, (4) studies not reporting tumor or treatment-related outcomes, (5) studies not providing comparative data between pre-pandemic and pandemic patient cohorts or comparing different patient populations, and (6) studies including patients with pathologies other than colorectal cancer.

### 2.1. Data Extraction and Outcomes Evaluated and Definitions

Data from included studies were extracted by two authors (GB, AP) and were entered into standardized excel spreadsheets (Microsoft, Redmond, Washington, DC, USA) for data tabulation. Data of primary importance were the time interval of data collection for the pre-pandemic and pandemic cohorts, the the tumor, node, and metastasis (TNM) classification (TNM) stage (AJCC 8th edition) [12] of the involved colorectal tumors, the number of total patients, the emergency presentation rates, neoadjuvant utilization rates, minimally invasive technique utilization rates, palliative-intent surgery rates, mortality/morbidity rates, and the length of hospital stay.

Secondary data were the patient demographics, the location of the tumors, the rates of stoma formation, the number of tumors complicated by obstruction or perforation, and the number of lymph nodes retrieved by surgery.

Minimally invasive surgery is defined as either laparoscopic or robotic abdominal surgery. Palliative intent surgery refers to palliative ostomy or by-pass procedures in cases in which tumor removal was either not possible or contraindicated. Tumors complicated by obstruction, perforation, or acute bleeding are referred to as complicated tumors throughout this analysis. Finally, stoma formation corresponds to the number of ostomies performed either for palliative reasons or for intentional diversion in cases of rectal surgery.

The present study was registered in the “International Prospective Register of Systematic Review” in 2021 (PROSPERO ID: CRD42021288432) and was conducted according to PRISMA guidelines [13].

### 2.2. Methodological Quality Assessment

Assessment of included studies for methodological integrity and accuracy of data reporting was performed using the Newcastle–Ottawa scale (NOS). The NOS is an 8-item scale evaluating the adequacy of the patient selection process (score of 0 to 4 stars), the comparability of the involved groups (0 to 2 stars), and the ascertainment of the reported exposure (0 to 3 stars). Each study is awarded a score of 0 to 9 stars, quantifying its methodological quality, with studies scoring 7 to 9 being of high quality, studies with scores from 5 to 6 being of mediocre quality, and studies with a score of 4 or less being of poor quality.

### 2.3. Statistical Analysis

Odds Ratios (OR) were calculated for the pooled analysis of dichotomous outcomes and Weighted Mean Differences (WMD) for continuous outcomes. The random-effects model was a priori selected to calculate ORs, WMDs, 95% Confidence Intervals (CI), and relevant *p*-values due to expected clinical heterogeneity in terms of geography, regional covid prevalence, and the extent of elective service disruption by Covid. The Higgin’s I^2^ statistic and relevant *p*-values were calculated to assess existing statistical heterogeneity between the sampled studies. All analyses were performed with the Revman v 5.4.1 (The Cochrane Collaboration, 2020) software. Funnel plots were constructed for each outcome (Appendix A); however, statistical testing for publication bias was not possible due to the small number of included studies. All *p*-values less than 0.05 were considered statistically significant.

Leave-one-out sensitivity analysis was performed for those outcomes in which the Higgin’s I^2^ test for heterogeneity was statistically significant or demonstrated a statistically significant effect size. The analysis was performed by iteratively removing one study at a time in order to identify outliers that contributed to the increased heterogeneity and evaluate the robustness of the analysis after the outliers were removed.

## 3. Results

After screening 283 abstracts generated by the search algorithm, 10 studies were deemed eligible for inclusion in the final quantitative analysis (Figure 1). In total, 26,808 patients were incorporated in the analysis (19,152 in the pre-pandemic cohort and 7656 in the pandemic cohort), with five studies being from east Asia (one from Japan [14], two from China [15,16], and two from Korea [17,18]) and the remaining five from Europe (two from the United Kingdom [11,19], one from Italy [20], one from Ireland [3], and one from Serbia [21]). The size of the pandemic cohorts in each study was compared to that of the pre-pandemic cohorts after matching for the time duration of the data collection in the pandemic cohorts (in months).

When analyzed for methodological quality, eight studies had an NOS score of 7 to 9 and were deemed of high methodological quality, and two studies had a score of 5 or 6 and were deemed of mediocre methodological quality. The median score of the obtained NOS scores was 7.5 (Table 1).

### 3.1. Tumor-Related Factors

Overall, the T3 stage was the most commonly encountered, both in the pre-pandemic (14.5%) and the pandemic (20%) cohorts. No statistically significant differences were encountered between the compared patient populations regarding the tumor extent (T) or nodal (N) stages of the involved tumors (Table 2). On the contrary, the number of patients presenting with metastases was found to be significantly increased in the pandemic cohort (OR 1.65, 95% CI 1.02–2.67, *p* = 0.04), with high interstudy heterogeneity (I^2^ = 91%).

Data on the distribution of tumors throughout the hindgut were available in eight studies, reporting similar patterns of distribution in both the pandemic and the pre-pandemic cohorts, with right-sided tumors (of the cecum, ascending, and transverse colons) being the most common tumor type in both the pre-pandemic (37.5%) and the pandemic (37.4%) cohorts. Forest plots are available for review in the Appendix A.

### 3.2. Presentation-Related Factors

Data from three studies encompassing 18,965 patients indicated a statistically significant increase of emergency presentations during the pandemic (OR 1.74, 95% CI 1.07–2.84, *p* = 0.03, Appendix A), from 19.3% in the pre-pandemic group of patients to 27.3% in the pandemic group. Study heterogeneity was high (I^2^ = 95%). Although there was a trend towards increased number of complicated tumors (by perforation or obstruction) in the pandemic group of patients (5.1% versus 3.9% in the pre-pandemic group), the difference was not statistically significant (Table 2, Appendix A).

### 3.3. Treatment-Related Factors

Neoadjuvant therapy utilization rates were significantly higher during the pandemic era, administered to 30.6% of the group population versus 27.4% of the group population during the pre-pandemic era (OR 1.22, 95% CI 1.09–1.37, *p* < 0.001, Appendix A), with no encountered interstudy heterogeneity (I^2^ = 0%). Similarly, patients in the pandemic group were more likely to be treated with palliative intent (OR 1.95, 95% CI 1.13–3.36, *p* = 0.02, I^2^ = 54%, Appendix A).

Minimally invasive approaches were more commonly used in the pre-pandemic group (48.1% versus 34.9% in the pandemic group, Appendix A), albeit without attaining statistical significance (*p* = 0.2). The same was evident for the stoma formation rates as well, with the pre-pandemic stoma rates being 10.2% and the pandemic being 8.4% (*p* = 0.74, Table 2, Appendix A).

### 3.4. Treatment Outcome Factors

There was no statistically significant difference recorded in terms of length of hospital stay (*p* = 0.49, Appendix A), lymph node yield (*p* = 0.39, Appendix A), or postoperative morbidity (*p* = 0.76, Appendix A). Mortality rates were only reported in a single study on 13,060 patients, indicating mortality rates of 1.6% in the pre-pandemic patient population and 2.6% in the pandemic one (Table 2).

### 3.5. Sensitivity Analysis

No single study was found to be the cause of the increased statistical heterogeneity observed for the T3/T4 stage, right-sided, left-sided, rectal tumor location, complicated tumor presentation, palliative intent surgery, minimally invasive surgery, and stoma formation outcomes. The study by Shinkwin et al. [11] was responsible for the encountered heterogeneity in the metastatic tumor (M+) outcome, while the study by Lim et al. [18] was responsible for the observed heterogeneity in the emergency presentation outcome. In both cases, after removal of the outlier studies, the recalculated OR failed to retain their statistical significance, therefore suggesting the presence of a type I statistical error. Finally, the heterogeneity encountered in the length of hospital stay outcome was attributed solely to the study by Cui et al. [16]; however, removal of the study, the significance of the obtained OR, was not influenced.

## 4. Discussion

The COVID-19 pandemic has significantly decreased the availability of health services for non-covid patients [22]. With regard to colorectal neoplasias, the unprecedented burden of the pandemic on healthcare systems has led to a considerable reduction in the number of diagnostic and surveillance endoscopic procedures performed in the general population [23,24]. Recent reports indicate that such drastic changes in clinical practice are projected to lead to increased incidence and tumor stage upshifting as missed cancer cases keep accumulating, in turn increasing colorectal cancer-associated mortality [25]. The problem is further compounded by the encountered delays and postponements of elective colorectal surgical procedures [2], carrying severe implications regarding the long-term impact of the pandemic in the survival of colorectal cancer patients.

Patient demographics and location of the tumors in the pandemic group of patients did not exhibit any significant differences when compared to the pre-pandemic group. Similarly, no differences were registered regarding the T and N staging of involved tumors; however, there was a significant increase in the number of patients presenting with metastatic neoplasms during the pandemic. In fact, patients operated for colorectal cancer during the pandemic were 65% more likely to be affected by metastatic colorectal tumors (*p* = 0.04). This implies that diagnostic and treatment delays may have led to significant tumor upstaging as has been previously postulated [26,27]. Although this explanation is both worrying and compelling, it should be examined in the context of reduced elective, but not emergent, operations during the pandemic era.

More specifically, results obtained from the present meta-analysis demonstrate that the odds of performing an emergency operation were significantly increased (by 74%, *p* = 0.03) during the pandemic, similarly to the odds for palliative-intent surgery (by 95%, *p* = 0.02). Both of these findings suggest that patients with advanced, symptomatic tumors comprised a larger percentage of the patient pool treated surgically during the pandemic, indicating that surgical healthcare accessibility was preferentially maintained for this particular patient subset. However, when patients with complicated tumors (by perforation or obstruction) were assessed, no statistically significant differences were encountered (Table 1, *p* = 0.18), although it is plausible that this is the result of a type II statistical error due to the small number of patients involved in this outcome. 

Another finding of the pooled data analysis is the increased rate of neoadjuvant therapy utilization. Patients receiving surgical treatment for colorectal cancer during the pandemic were 22% more likely to have undergone neoadjuvant therapy (*p* < 0.001). Deferral of surgery in favor of neoadjuvant therapy appears to provide an effective solution while waiting for the resumption of normal elective surgical practice. Morris et al. in a population-based study in England reported a 44% increase in the rate of neoadjuvant therapy utilization for rectal cancer during the pandemic era, with long-course regimens being preferred over short-course ones [2]. In our pooled patient cohort, neoadjuvant therapy rates rose form 27.4% in the pre-pandemic group to 30.6% in the pandemic group, clearly indicating a shift in clinical practice brought upon by the COVID-19 pandemic, which appears to be consistent across different included studies (I^2^ = 0%) and falls within the general underlying trend favoring surgery deferral.

Surgical practices did not significantly differ, although there was a clear trend towards less minimally invasive surgery in the pandemic cohort (34.8% of cases vs. 48.1% in the pre-pandemic cohort). This change in practice is directly attributable to fear of COVID-19 spread via aerosolization during pneumoperitoneum evacuation, a concern that has controversially led to opposition of minimally invasive techniques during the initial phase of the pandemic [28,29]. Subsequently published guidelines from surgical societies have proposed reinstatement of minimally invasive techniques in both elective and emergency surgical practice, provided that precautionary protective practices, such as use of personal protective equipment, pneumoperitoneum release filters, and liberal preoperative COVID-19 molecular testing are followed [30,31]. Despite the reduced rates of minimally invasive technique utilization in the pandemic cohort, the length of hospital stay—although inconsistently reported in included studies—remained comparable between the two evaluated groups. The same holds true for stoma formation rates as well, despite the evidently increased rates of emergency surgery, suggesting that the employed surgical strategies remained roughly unchanged.

Lymph node yield, morbidity, and mortality rates could rationally represent metrics of surgical safety and efficacy; however, they were seldom reported. A eta-analysis of two studies [15,16] revealed a combined morbidity rate of 6.1% in pre-pandemic patients versus 4.8% in their pandemic counterparts, without any statistically significant difference. Similarly, lymph node yield was equivalent in the two groups based on pooled outcomes from three studies [16,20,21]. Although the paucity of data precludes any concrete assumptions to be made, safety of surgical practice appears to be maintained in the pandemic era despite the looming threat of COVID-19 infection, with oncologic surgical benchmarks being seemingly comparable to the pre-pandemic controls.

The study by Kuryba et al. [19] was the only study to report data on mortality, derived from a large UK-based population registry, revealing 2.6% mortality rates in the pandemic cohort versus 1.6% in the pre-pandemic one. The encountered marginal increase in mortality was mainly attributable to emergency surgery cases (OR 1.74, *p* = 0.003) and was especially pronounced in so-called “hot-sites”, i.e., centers that accommodated COVID-19 positive patients as well as elective and emergency colorectal cancer cases. More importantly, COVID-19 superinfection in patients undergoing surgery resulted in a tenfold increase in mortality rates in both elective and emergency cases and was accompanied by prolongation of hospital stay that reached a median of 17 to 20 days, highlighting the detrimental effects of COVID-19 infection in such patients.

One major limitation of the present study is that the evaluated outcomes were inconsistently reported amongst included studies reporting on widely varying number of patients as is exhibited in Table 2. Moreover, differences in the local prevalence of COVID-19, the type of confinement and social distancing measures imposed by governmental authorities, and the extent of surgical service disruption could not be accounted for, given that the majority of included studies are single-center reports. This fact renders the obtained results highly susceptible to the possibility of selection and sampling biases. Another caveat of the meta-analysis is the inability to assess the impact of the pandemic on disease-free and overall survival, which remain as yet uncertain. Finally, there was high interstudy statistical heterogeneity encountered for the metastatic tumors and emergency surgery outcomes, coupled together with the results of the sensitivity analysis that demonstrated the presence of single study outliers, whose subsequent removal altered the significance of the obtained cumulative results. As such, no concrete conclusions can be reached for these particular outcomes, and they should be interpreted with caution.

## 5. Conclusions

The COVID-19 pandemic has resulted in demonstrable changes in the surgical practice involving colorectal malignancies. Patients during the pandemic were more likely to undergo palliative interventions as opposed to pre-pandemic controls. In addition, neoadjuvant therapy utilization rates were increased in the pandemic era as a means of safe postponement of surgery in select cases. These evident changes in practice recorded during the initial months of the pandemic are rational, taking into account the severe sequelae of COVID-19 infection during the perioperative period. However, they may be alarmingly associated with tumor upstaging, with significant implications regarding the long-term oncologic survival of patients with colorectal neoplasias.

## Figures and Tables

**Figure 1 cancers-14-01229-f001:**
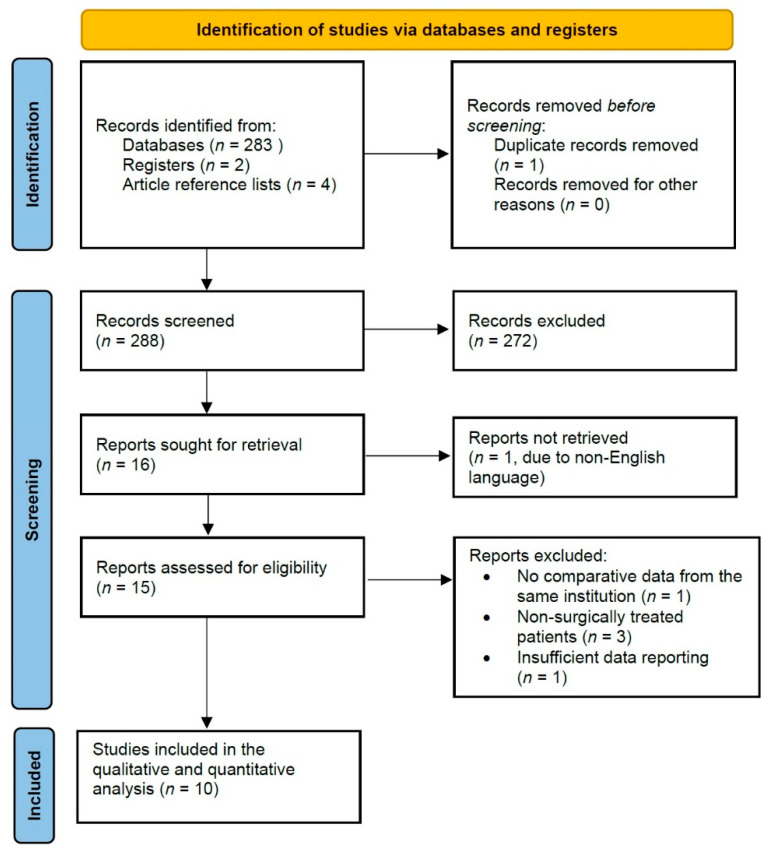
Prisma flowchart of study selection.

**Table 1 cancers-14-01229-t001:** Included study characteristics.

Study	Country	Time Interval of Data Collection	Total Patients	Age (Mean ± SD)	Sex (Male/Female)	Newcastle–Ottawa Scale Score
Pre-pandemic versus Pandemic cohorts, *n* (%)						
Donlon [3]	Ireland	March 2019–March 2020 vs. March 2020–March 2021	1631 vs. 1093	N/a	N/a	7
Peltrini [20]	Italy	October 2019–February 2020 vs. January 2021–May 2021	41 vs. 43	N/a	N/a	5
Lim [18]	Korea	January–July 2017–2019 vs. January–July 2020	2514 vs. 715	61 (18–90) vs. 61 (17–97) *	1484 (59%)/1030 (41%) vs. 415 (58%)/300(42%)	8
Shinkwin [11]	UK	January–December /2018–2019 vs. January–December 2020	539 vs. 267	70 ± 12.5 vs. 70 ± 14	308 (57.1%)/231 (42.9%) vs. 151 (56.6%)/116 (43.4%)	8
Kuryba [19]	UK	Six weeks before 23 March 2020 vs. nine weeks after	11703 vs. 3227	N/a	6586 (56.2%)/5117 (43.8%) vs. 1793 (55.5%)/1434 (44.5%)	9
Choi [17]	Korea	March-September 2018–2019 vs. March-September 2020	1985 vs. 916	62.6 ± 12.2 vs. 61.7 ± 12.1	1160 (58.4%)/825 (41.6%) vs. 524 (57.2%)/392 (42.8%)	9
Radulovic [21]	Serbia	January–December 2019 vs. March 2020–April 2021	152 vs. 49	67.11 ± 11.62 vs. 67.41 ± 10.37	87 (57.2%)/65 (42.8%) vs. 22 (44.9%)/27 (55.1%)	6
Xu [15]	China	January–May 2019 vs. January–May 2020	828 vs. 710	N/a	518 (62.6%)/310 (37.4%) vs. 438 (61.7%)/272 (38.3%)	7
Cui [16]	China	February-May 2018–2019 vs. February-May 2020	205 vs. 67	65.6 ± 11.65 vs. 67.1 ± 11.4	111 (54.1%)/94 (45.9%) vs. 44 (65.7%)/23 (34.3%)	9
Mizuno [14]	Japan	December 2018–April 2020 vs. April 2020–August 2020	92 vs. 31	72.91 ± 10.58 vs. 72 ± 10.7	54 (58.7%)/38 (51.3%) vs. 25 (80.6%)/16 (19.4%)	9

N/a = Not available. * Data presented as median (range).

**Table 2 cancers-14-01229-t002:** Pooled analysis outcomes.

Outcome	Number of Studies	Total Patients	Patients in the Prepandemic Cohort *n*(%)	Patients in the Pandemic Cohort *n*(%)	OR/WMD	95% Confidence Intervals	*p*-Value	I^2^	I^2^ *p*-Value
Tumor-Related Factors									
Tis-T1 stage	5	7301	628 (4.1)	276 (6.3)	1.14	0.87–1.48	0.34	41%	0.15
T2 stage	5	7301	703 (4.6)	255 (5.9)	0.91	0.78–1.06	0.2	0%	0.6
T3 stage	5	7301	2198 (14.5)	883 (20)	1.18	0.82–1.7	0.38	88%	<0.001
T4 stage	6	7385	736 (4.2)	290 (5.7)	1.19	0.79–1.8	0.4	80%	<0.001
N + stage	6	7385	1797 (10.2)	720 (14.3)	1	0.89–1.11	0.96	0%	0.54
M + stage	6	19,414	2020 (11.8)	711 (13.5)	1.65	1.02–2.67	0.04	91%	<0.001
Right-sided tumors	7	19,893	5294 (37.5)	1834 (37.4)	0.88	0.51–1.52	0.66	99%	<0.001
Left-sided tumors	7	19,893	4946 (35)	1759 (35.9)	0.91	0.56–1.5	0.72	96%	<0.001
Rectal tumors	8	22,794	4794 (29.8)	1934 (33.2)	0.93	0.63–1.37	0.71	95%	<0.001
Presentation-Related Factors									
Emergency presentation	3	18,965	2851 (19.3)	1149 (27.3)	1.74	1.07–2.84	0.03	95%	<0.001
Complicated tumor	3	4562	113 (3.9)	84 (5.1)	1.72	0.78–3.78	0.18	82%	0.004
Treatment-Related Factors									
Neoadjuvant therapy	3	7668	1459 (27.4)	656 (30.6)	1.22	1.09–1.37	<0.001	0%	0.4
Palliative intent surgery	4	4795	114 (6.6)	126 (7.3)	1.95	1.13–3.36	0.02	54%	0.09
Minimally Invasive Surgery	6	22,584	7680 (48.1)	2056 (34.9)	0.68	0.37–1.24	0.2	98%	<0.001
Stoma Formation	5	19,683	1425 (10.2)	479 (8.4)	0.91	0.51–1.62	0.74	94%	<0.001
Treatment Outcome Factors									
Mortality	1	13,060	163 (1.6)	74 (2.6)	N/a	N/a	N/a	N/a	N/a
Morbidity	2	1810	63 (6.1)	37 (4.8)	0.92	0.55–1.55	0.76	25%	0.25
Length of hospital stay	3	2011	N/a	N/a	0.51	−0.93–1.94	0.49	79%	0.008
Lymph node harvest	3	1894	N/a	N/a	1.57	−1.99–5.13	0.39	64%	0.06

Tis = T in situ, OR= Odds Ratio, WMD = Weighted Mean Difference, I^2^ = Higgin’s I^2^ statistic, N/a = Not available.

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
