# Peer review of "Colorectal Surgery in the COVID-19 Era: A Systematic Review and Meta-Analysis"

_cancers, 2022, doi:10.3390/cancers14051229_

Round 1

Reviewer 1 Report

Dear authors,

Thanks for revising the manuscript. I still have some minor questions.

  • Regarding your statement;

Following the reviewer’s suggestion we have changed the text to: “Eight studies recorded a reduction in the total caseload during the pandemic, ranging from 7.7% to 72.4%, while the studies by Peltrini et al. and Shinkwin et al. demonstrated only marginal deviations from the pre-pandemic controls (4.8% in-crease and 1% decrease in the total case volume respectively, Table 1)”.

For Peltrini et al, please add actual numbers in the text as 4.8% seems to be a deviation.

  • About Shinkwin et al; you wrote that interval of data collection in the pre-pandemic group was double (two months) that the one in the pandemic group (one month). But in the table 1, you have entered - 01-12/2018 and 2019 vs 01-12/20

Is it 2 years vs one year (am I misinterpreting it?)

  • Regarding my comment about figure 1 and registers;

Title of Figure 1 is identification of studies via databases and registers.

Please elaborate in the methods what registers did you use as you have written, “A systematic literature search of the Medline, Scopus, Web of Science, and China National Knowledge Infrastructure (CNKI) and clinicaltrials.gov databases………” No mention of any register? Reader may get confused.

  • About “Just curious about one report that was not retrieved.”

Add the reason for not retrieving.

  • Regarding, “In Discussion: Briefly discuss studies Like Peltrini/ Shinkwin showing the differences between two cohorts in presentation even when the numbers are not much different.”

Please elaborate the reasons, why these two studies didn’t differentiate in two periods, pre-pandemic time and during pandemic. It could be due to the reasons as you pointed out in your limitation statement, “inconsistently reported amongst included studies incorporating widely varying number of patients as is exhibited on Table 2.” Or  you can review those papers again and report what reasons or limitation these two studies have given for the outcomes we have seen – marginal difference in two time points.

By the way, you have misspelled the word- “incrorporating” in this statement.

  • Regarding Kuryba et al, in the discussion section (page 9) p value is still incorrect. Please enter correct information. (OR 1.74, p=003)
  • Table 2: It would be good if you also provide the breakdown (into pre-pandemic and pandemic) of the total number of patients for each row. Otherwise, it is useless for each row to have total number of patients as it does not make sense without the breakdown. How did you calculate percentage for pandemic and pre-pandemic cohorts in this table?

Author Response

Response to Reviewer #1:

We sincerely thank the reviewer for his/her time taken reviewing our manuscript and for the insightful comments. Our response follows (the reviewer’s comments are in Italics):

  1. Regarding your statement; Following the reviewer’s suggestion we have changed the text to: “Eight studies recorded a reduction in the total caseload during the pandemic, ranging from 7.7% to 72.4%, while the studies by Peltrini et al. and Shinkwin et al. demonstrated only marginal deviations from the pre-pandemic controls (4.8% in-crease and 1% decrease in the total case volume respectively, Table 1)”. For Peltrini et al, please add actual numbers in the text as 4.8% seems to be a deviation.

Response: After carefully considering the reviewer’s concerns about possible confusion pertaining to our method used to calculate the average matched differences in caseload before and after the pandemic, we have decided to remove it altogether. We agree with the reviewer that this analysis is problematic and possibly misleading and have thus removed all its aspects from the manuscript.

  1. About Shinkwin et al; you wrote that interval of data collection in the pre-pandemic group was double (two months) that the one in the pandemic group (one month). But in the table 1, you have entered - 01-12/2018 and 2019 vs 01-12/20 Is it 2 years vs one year (am I misinterpreting it?)

Response: Following on from comment #1 and the previous discussion, this entire part of the results section, along with the relevant column on Table 1, were entirely removed to avoid misconceptions. Concerning our previous response, the interval of data collection in the pre-pandemic group was indeed double than the one in the pandemic group. The former was two years (not two months, this was a typo) and the latter one year. The respective cell on Table 1 was fixed to be uniform to the others, now displaying “01-12/2018-2019 vs 01-12/2020”. This is an abbreviated version of “01-12/2018 and 01-12/2019 vs 01-12/2020”.

  1. Regarding my comment about figure 1 and registers; Title of Figure 1 is identification of studies via databases and registers. Please elaborate in the methods what registers did you use as you have written, “A systematic literature search of the Medline, Scopus, Web of Science, and China National Knowledge Infrastructure (CNKI) and clinicaltrials.gov databases………” No mention of any register? Reader may get confused.

Response: We thank the reviewer for pointing this out. As mentioned in our previous response, only one register was used, clinicaltrials.gov (please note that the Registers, n=2 on Figure 1 refers to the number of studies identified through registers, not the number of registers used). We have adjusted the methods text to state that: “A systematic literature search of the MedLine, Scopus, Web of Science, and China National Knowledge Infrastructure (CNKI) databases and clinicaltrials.gov register…”. We sincerely hope that this clears any confusion on our employed literature search methods.

  1. About “Just curious about one report that was not retrieved.” Add the reason for not retrieving.

Response: The study was irretrievable due to the publisher being Chinese and thus the full-text could not be accessed. This was added on Figure 1 (the PRISMA flowchart).

  1. Regarding, “In Discussion: Briefly discuss studies Like Peltrini/ Shinkwin showing the differences between two cohorts in presentation even when the numbers are not much different.” Please elaborate the reasons, why these two studies didn’t differentiate in two periods, pre-pandemic time and during pandemic. It could be due to the reasons as you pointed out in your limitation statement, “inconsistently reported amongst included studies incorporating widely varying number of patients as is exhibited on Table 2.” Or you can review those papers again and report what reasons or limitation these two studies have given for the outcomes we have seen – marginal difference in two time points.

Response: We thank the reviewer for further explaining his/her comment. Upon reviewing these two studies, no particular reason for the marginal (almost non-existent) change in the caseload of colorectal cancer patients. It stands to logic that some centers experienced only small derangements in their clinical practice during the course of the pandemic. The magnitude of service derangement likely depended on local health policies, the prevalence of the coronavirus in the local population and the time period during which the data collection took place (all of which are stated in the limitations section of the discussion). Following on from the responses to comments #2 and #3, we believe that the reviewer will find it acceptable that we choose not to individually discuss these studies in the revised version of the manuscript, given that we entirely removed the percent difference in total patient caseload before and after the pandemic.

  1. By the way, you have misspelled the word- “incrorporating” in this statement.

Response: Thank you for pointing this out. The statement was changed to “were inconsistently reported amongst included studies reporting on widely varying number of patients”

  1. Regarding Kuryba et al, in the discussion section (page 9) p value is still incorrect. Please enter correct information. (OR 1.74, p=003)

Response: Once again, thank you for pointing this out. The mistake was fixed (p=0.003).

  1. Table 2: It would be good if you also provide the breakdown (into pre-pandemic and pandemic) of the total number of patients for each row. Otherwise, it is useless for each row to have total number of patients as it does not make sense without the breakdown. How did you calculate percentage for pandemic and pre-pandemic cohorts in this table?

Response: We do acknowledge that our attempt to quantify the magnitude of patient caseload before and after the pandemic was misled. We theorized (based on our own clinical experience) that the pandemic caused reduction in the number of surgically treated patients with colorectal cancer, however, we now understand that the analysis presented in the current manuscript is problematic and cannot thus support such a hypothesis. The percentage change for the pandemic and the pre-pandemic cohorts was calculated after comparing the mean number of patients treated in the pre-pandemic cohort (oftentimes including multiple time periods, each one including a different number of patients) to the number of patients treated during the pandemic. We recognize that the attempt to forcefully describe a caseload reduction brought upon by the pandemic (whether true or not) could not be supported by the results of such a complicated and possibly biased analysis and finally elected to remove it from the manuscript.

Reviewer 2 Report

Summary

Pararas et al. performed a systematic review and meta-analysis to reveal the status of colorectal surgery in a pre-and post-COVID-19 era. Unfortunately, the authors did not correct the following points which I have mentioned the last version of review report.

Materials and Methods

1) In the searching criteria, Pubmed, Medline, and Google scholar have not been included. Many meta-analyses were performed with the results from these databases. Please consider including these databases for primary data search (van Doorn et al. Aliment Pharmacol Ther. 2020 Aug 27 : 10.1111/apt.16036.; Cheung et al. Neuropsychol Rev. 2016 Jun;26(2):121-8. doi: 10.1007/s11065-016-9319-z).

2.3. Statistical analysis

2) In this study the primary outcome and secondary outcomes had not been defined before the meta-analysis was performed. The authors should set the primary and secondary outcomes before the meta-analysis will be performed (Cheung et al. Neuropsychol Rev. 2016 Jun;26(2):121-8. doi: 10.1007/s11065-016-9319-z).

Results

3) In this study, the outcomes of M stage difference Emergency presentation and Palliative intent surgery only with this data have large heterogeneities (I2=54-95%). The authors should be careful about the interpretation of the results when these heterogeneities have been observed. I strongly recommend performing meta-regression analysis, sub-group analysis, or making forest plots and showing which studies contributed to these heterogeneities.

As the authors stated, this study contained large heterogeneity in each point which was evaluated by the meta-analysis, there is no conclusion should be written at this point. Thus, I cannot recommend to publish this article for this time.

Author Response

Response to Reviewer #2:

We sincerely thank the reviewer for his/her thorough reading of our manuscript. We are sorry that we failed to address the reviewer’s questions. We believe that this might stem from miscommunication. We will endeavor to reply to the reviewer’s comments as accurately as possible. Our response follows (the reviewer’s comments are in Italics):

Materials and Methods

  • In the searching criteria, Pubmed, Medline, and Google scholar have not been included. Many meta-analyses were performed with the results from these databases. Please consider including these databases for primary data search (van Doorn et al. Aliment Pharmacol Ther. 2020 Aug 27 : 10.1111/apt.16036.; Cheung et al. Neuropsychol Rev. 2016 Jun;26(2):121-8. doi: 10.1007/s11065-016-9319-z).

Response: In the Materials and Methods section of the manuscript, we clearly state searching the Medline (i.e. Pubmed) databases, along with others such as the Scopus, Web of Science, and China National Knowledge Infrastructure (CNKI) databases were used to identify studies. Google scholar was not utilized as per the reasoning described in our previous response.

Statistical analysis

  • In this study the primary outcome and secondary outcomes had not been defined before the meta-analysis was performed. The authors should set the primary and secondary outcomes before the meta-analysis will be performed (Cheung et al. Neuropsychol Rev. 2016 Jun;26(2):121-8. doi: 10.1007/s11065-016-9319-z).

Response: The primary and secondary outcomes of the study were determined and registered before the meta-analysis was conducted, even before the literature search was completed. The predetermined protocol of the study can be accessed here:

https://www.crd.york.ac.uk/prospero/display_record.php?RecordID=288432

Results

  • In this study, the outcomes of M stage difference Emergency presentation and Palliative intent surgery only with this data have large heterogeneities (I2=54-95%). The authors should be careful about the interpretation of the results when these heterogeneities have been observed. I strongly recommend performing meta-regression analysis, sub-group analysis, or making forest plots and showing which studies contributed to these heterogeneities.

Response: We wholly agree that results with large heterogeneities should be interpreted with caution. In fact, we forcefully state this at the limitations section of the discussion (“Finally, the high interstudy statistical heterogeneity encountered for the metastatic tumors and emergency surgery outcomes, coupled together with the results of the sensitivity analysis that demonstrated the presence of single study outliers, whose subsequent removal altered the significance of the obtained cumulative results. As such, no concrete conclusions can be reached for these particular outcomes and they should be interpreted with caution”.

Exploratory analyses for the increased heterogeneity (such as meta-regression analysis and sub-group analysis) could not unfortunately be performed due to literature constraints. We did include however a leave-one-out sensitivity analysis to hopefully add more insight on the sources of the encountered heterogeneity. We also added the forest plots as supplementary material for completeness’ sake.

  • As the authors stated, this study contained large heterogeneity in each point which was evaluated by the meta-analysis, there is no conclusion should be written at this point. Thus,I cannot recommend to publish this article for this time.

Response: Both heterogeneity and the possibility of selection bias are high in this meta-analysis for many of the outcomes as accurately stated by the reviewer. This problem is inherent to type of data utilized and not of the methods used. This is clearly stated as a limitation of the present meta-analysis, however, it does not negate the results obtained by the pooled data analysis, which in our opinion are an accurate reflection of the existing literature on the topic.

We are sorry that we could not address this particular concern of the reviewer, nevertheless we are thankful for his/her comments and corrections, most of which have already been incorporated into our manuscript during our previous response.

Round 2

Reviewer 2 Report

The manuscript has been revised well and I agree that the manuscript described the significant impact of the COVID-19 on the treatment of colorectal surgery. However, there are still major concerns and I cannot recommend publishing this manuscript for now.

1) The authors performed the sensitivity analysis to reveal the cause of large heterogeneity on metastatic tumor (M+) outcome and emergency presentation outcome. The authors then stated that after removal of the Shinkwin’s and Lim’s studies, the Odds ratios (ORs) are not statistically significant in both analyses. The authors discussed this was due to the type I error. However, considering the 95% confidence interval (CI) were 1.02-2.67 in the M+ stage and 1.07-2.84 in an emergency presentation in this study, it is more reasonable to conclude that the COVID-19 pandemic had an impact on the length of hospital stay, and it is not determinable whether the patients were more likely to present with metastatic tumors and emergency surgery. I would concern that the sample sizes are enough to evaluate OR after the removal of these studies (Imberger et al. BMJ Open. 2016; 6(8): e011890.).

I also concerned about the heterogeneity in the palliative intent surgery (I2=54%) in the previous reviewer’s report. The authors did not mention this heterogeneity in the main text. Please reconsider revising the main text.

Round 3

Reviewer 2 Report

This paper is a significant contribution, and I think the current revision can be accepted for publication.

This manuscript is a resubmission of an earlier submission. The following is a list of the peer review reports and author responses from that submission.

Round 1

Reviewer 1 Report

Nikolaos Pararas et al in their manuscript entitled, “Colorectal Surgery in the Covid-19 Era: A Systematic Review and Meta-Analysis” presents an interesting review study that evaluated the impact of Covid-19 pandemic on the surgical treatment of colorectal cancer. The decrease of routine or non-emergency hospital visits led to missed diagnosis and presentation of cases at a much-advanced stage. The COVID-19 pandemic has created a global medical crisis affecting all cancers including colorectal cancer.

The manuscript is well written and conducted a meta-analysis of 10 studies by providing a systemic review for the medical community regarding the influence of pandemic on untimely treatment or delayed presentation of CRC. This may have an impact on the potentially increased mortality due to colorectal cancer.

I have the following suggestion for the authors to improve the manuscript.

In the introduction or discussion, please briefly discuss the role of delayed colorectal screening causing the tumor upstaging or increased complication at the time of presentation. (You can cite a recently published study: (https://doi.org/10.3390/life11121297) in addition to references #7, 22, and 23.

I have some additional comments.

  • Statistical analysis:

Please explain existing statistical heterogeneity between the sampled studies. Are you referring to all the studies or studies presented in table 2) for certain parameters.

  • Some of the text is in bold or with Font with increased size, please fix it. “(The Cochrane Collaboration, 2020) software. Assessment for publication bias with funnel plots was not possible due to the small number of included studies.”
  • Results and abstract:
  • You should not rephrase this differently “Nine studies recorded a reduction in the total caseload during the pandemic, ranging from 1% to 72.4%, while a single study recorded a 4.8% increase in the total case volume (Table 1)”. The Peltrini study has only 41 patients before pandemic and 43 during pandemic so the number is low or rather not much different but presenting as an increase of 4.8% seems much bigger. Similarly, the Shinkwin study has only a -1% difference although the overall patient numbers were high in both cohorts.

I would say you should write on the lines that 8 studies evidenced decrease (provide a range) and two studies showed almost no difference (provides their actual numbers from 41 vs 43 and 539 vs 267). About Shinkwin study, how did you get this -1%? You can also point out that about the disease staging of Peltrini study patients.

  • Figure 1: You mentioned using 2 registers (I think only one is cited in the methods - PROSPERO) I might have missed it.

Just curious about one report that was not retrieved.

Studies included n=10/ reports included n=10, it seems like you have 20 different sources of this data. Please clarify.

  • Table 1: It would be good to add a citation number (as superscript) for each study title in the table.

For the column "Age", please mention that it is mean (SD) except Lim’s study.

  • Table 2: Please add a footnote and explain, Tis-T1stage (for better understanding), I2 and I2 p value.

It would be good if you also provide the breakdown (into pre-pandemic and pandemic) of the total number of patients for each row.

  • On page 3, line correct; (37,4%)
  • Discussion: Briefly discuss studies Like Peltrini/ Shinkwin showing the differences between two cohorts in presentation even when the numbers are not much different.

In the discussion section, your statement is confusing while discussing a study by Kuryba et al.

The data presented here do not match (or I may not understand) with the data shown in “Table 2 - Treatment outcome  - Mortality”

It is 1.6 vs 2.6 in the Table as compared to written here, “the only study to report data on mortality, derived from a large UK-based population registry, revealing 1.6% mortality rates in the pandemic cohort versus 2.6% in the pre-pandemic one. The encountered marginal increase in mortality was mainly attributable to emergency surgery cases (OR 1.74, p=0.003).” Even the p value in table 2 is 0.03 for emergency presentation.

Reviewer 2 Report

Summary

Pararas et al. performed a systematic review and meta-analysis to reveal the status of colorectal surgery in a pre-and post-COVID-19 era. Although the authors performed a meta-analysis well, I cannot recommend this article for publication because it has extensive problems.

Major points

Materials and Methods

1) In the searching criteria, Pubmed, Medline, and Google scholar have not been included. Many meta-analyses were performed with the results from these databases. Please consider including these databases for primary data search (van Doorn et al. Aliment Pharmacol Ther. 2020 Aug 27 : 10.1111/apt.16036.; Cheung et al. Neuropsychol Rev. 2016 Jun;26(2):121-8. doi: 10.1007/s11065-016-9319-z).

2.1. Data extraction and outcomes evaluated

2) The definition of “minimally invasive surgery” is not clear (e.g. Laparoscopic and/or Robotic surgery). Please clarify the definition of  “minimally invasive surgery” (Kryzauskas et al. BMC Geriatr. 2021 Dec 7;21(1):682. doi: 10.1186/s12877-021-02648-2.).

3) It is not clear if the term “palliative-intent surgery” includes endoscopic stenting or not. Please clarify the definition of this term and which “palliative” technique is included in this analysis.

4) The reference of TNM staging is not clear. Please describe which TNM factor has been used for this study.

2.3. Statistical analysis

5) In this study the primary outcome and secondary outcomes had not been defined before the meta-analysis was performed. The authors should set the primary and secondary outcomes before the meta-analysis will be performed (Cheung et al. Neuropsychol Rev. 2016 Jun;26(2):121-8. doi: 10.1007/s11065-016-9319-z).

6) Although the authors stated that the assessment for publication bias with funnel plots was not possible due to the small number of included studies, the authors should show the funnel plots so that readers can visually identify the publication biases.

7) In this study, there is no forest plot in any primary data. In the meta-analysis, the forest plot is important to see the heterogeneity and overall effect in each evaluating point. Please describe and show the forest plot (Rydzewska et al. Eur J Cancer. 2017 Oct; 84: 88–101.).

Results

8) In this study, the outcomes of M stage difference Emergency presentation and Palliative intent surgery only with this data have large heterogeneities (I2=54-95%). The authors should be careful about the interpretation of the results when these heterogeneities have been observed. I strongly recommend performing meta-regression analysis, sub-group analysis, or making forest plots and showing which studies contributed to these heterogeneities.